# Understanding the World Through Action

**Sergey Levine**
Berkeley AI Research (BAIR)
UC Berkeley
svlevine@eecs.berkeley.edu

**Abstract:** The recent history of machine learning research has taught us that machine learning methods can be most effective when they are provided with very large, high-capacity models, and trained on very large and diverse datasets. This has spurred the community to search for ways to remove any bottlenecks to scale. Often the foremost among such bottlenecks is the need for human effort, including the effort of curating and labeling datasets. As a result, considerable attention in recent years has been devoted to utilizing unlabeled data, which can be collected in vast quantities. However, some of the most widely used methods for training on such unlabeled data themselves require human-designed objective functions that must correlate in some meaningful way to downstream tasks. I will argue that a general, principled, and powerful framework for utilizing unlabeled data can be derived from reinforcement learning, using general purpose unsupervised or self-supervised reinforcement learning objectives in concert with offline reinforcement learning methods that can leverage large datasets. I will discuss how such a procedure is more closely aligned with potential downstream tasks, and how it could build on existing techniques that have been developed in recent years.

**Keywords:** Self-Supervised Learning, Embodied Learning

## 1 Introduction

Machine learning systems have mastered a breadth of challenging problems in domains ranging from computer vision [1] to speech recognition [2] and natural language processing [3], and yet the question of how to design learning-enable systems that match the flexibility and generality of human reasoning remains out of reach. This has prompted considerable discussion about what the "missing ingredient" in modern machine learning might be, with a number of hypotheses put forward as the big question that the field must resolve. Is the missing ingredient causal reasoning [4], inductive bias [5], better algorithms for self-supervised or unsupervised learning [6], or something else entirely?

This question is difficult one, and any answer must necessarily involve a great deal of guesswork, but the lessons we can glean from recent progress in artificial intelligence can provide us with several guiding principles.

The first lesson is in the "unreasonable" effectiveness of large, generic models supplied with large amounts of training data. As eloquently articulated by Sutton [7] in his essay on the "bitter lesson," as well as a number of other researchers in machine learning, a persistent theme in the recent history of ML research has been that methods that effectively leverage large amounts of computation and large amounts of data tend to often outperform methods that rely on manually engineered priors and heuristics. While a full discussion of the reasons for this trend are outside the scope of this article, in short they could be summarized (or perhaps caricatured) as follows: when we engineer biases or priors for our models, we are injecting our own imperfect knowledge of how the world works, which biases the model toward some solutions over others. When the model instead gleans this knowledge from data, it will arrive at conclusions that are more accurate than those that we engineered ourselves, and therefore will work better. Indeed, a similar pattern has been observed in how people acquire proficiency. As discussed by Dreyfus and Dreyfus [8], "rule-based" reasoning that follows rules we can articulate clearly tends to only provide people with "novice-level" performance at var-

Blue Sky Papers, 5th Conference on Robot Learning (CoRL 2021), London, UK.

ious skills, while "expert-level" performance is associated with a mess of special cases, exceptions, and patterns that people struggle to articulate clearly, and yet can leverage seamlessly in the moment when the situation demands it. As Dreyfus points out, real human experts are rarely able to articulate the rules they actually follow when exhibiting their expertise, and so it should come as no surprise that in the same way that we must acquire expertise from experience, so must our machines. And to do that, they will need powerful, high-capacity models that impose comparatively few biases and can handle the large amounts of experience that will be needed.

The second, more recent lesson is that manual labeling and supervision do not scale nearly as well as unsupervised or self-supervised learning. We've already seen that unsupervised pre-training has become standard in natural language processing [9], and perhaps will soon become standard across other fields. In a sense, this lesson is a corollary to the first one: if large models and large datasets are the most effective, then anything that limits how large the models and datasets can be will eventually end up as the bottleneck. Human supervision can be one such bottleneck: if all data must be labeled manually by a person, then less data will be available to the system to learn from. However, here we reach a conundrum: current methods for learning without human labels often violate the principles outlined in the first lesson, requiring considerable human insight (which is often domain-specific!) to engineer self-supervised learning objectives that allow large models to acquire meaningful knowledge from unlabeled datasets. These include relatively straightforward tasks such as language modeling [10], as well as comparatively more esoteric tasks, such as predicting whether two transformed images were produced by the same original image, or two different ones [11]. The latter is a widely used and successful approach in modern self-supervised learning in computer vision. While such approaches can be effective up to a point, it may well be that the next bottleneck we will face is in deciding how to train large models without requiring manual labeling *or* manual design of self-supervised objectives, so as to acquire models that distill a deep and meaningful understanding of the world and can perform downstream tasks with robustness generalization, and even a degree of common sense.

I will argue that such methodology could be developed out of current algorithms for learning-based control (reinforcement learning), though it will require a number of substantial algorithmic innovations that will allow such methods to go significantly beyond the kinds of problems they've been able to tackle so far. Central to this idea is the notion that, in order to control the environment in diverse and goal-directed ways, autonomous agents will necessarily need to develop an understanding of their environment that is causal and generalizable, and hence will address many of the shortcomings of current supervised models. At the same time, this will require going beyond the current paradigm in reinforcement learning in two important ways. First, reinforcement learning algorithms require a task goal (i.e., a reward function) to be specified by hand by the user, and then learn the behaviors necessary to accomplish that task goal. This of course greatly limits their ability to learn without human supervision. Second, reinforcement learning algorithms in common use today are not inherently *data-driven*, but rather learn from online experience, and although such methods can be deployed directly in real-world environments [12], online active data collection limits their generalization in such settings [13], and many uses of reinforcement learning instead take place in simulation, where there are few opportunities to learn about how the *real* world works.

## 2   Learning Through Action

Insofar as artificial intelligence systems are useful, it is because they provide inferences that can be used to make decisions, which in turn affect something in the world. Therefore, it is reasonable to conclude that a general learning objective should be one that provides impetus to learn those things that are most useful for affecting the world in meaningful ways. Making decisions that create desired outcomes is the purview of reinforcement learning and control. Therefore, we should consider how reinforcement learning can provide the sort of automated and principled objectives for training high-capacity models that can endow them with the ability to understand, reason, and generalize.

However, this will require addressing the two limitations: reinforcement learning requires manually-defined reward functions, and it requires an active learning paradigm that is difficult to reconcile with the need to train on large and diverse datasets. To address the issue with objectives, we can develop algorithms that, instead of aiming to perform a single user-specified task, rather aim to accomplish whatever outcomes they infer are possible in the world. Potential objectives for such methods could include learning to reach any feasible state, learning to maximize mutual information

between latent goals and outcomes, or learning through principled intrinsic motivation objectives that lead to broad coverage of possible outcomes. To address the issue with data, we must develop reinforcement learning algorithms that can effectively utilize previously collected datasets. These are offline reinforcement learning algorithms [13], and they can provide a path toward training RL systems on broad and diverse datasets in much the same manner as in supervised learning, followed by some amount of active online finetuning to attain the best performance [14].

To provide a hypothetical example of a system that instantiates these ideas, imagine a robot that performs a variety of manipulation tasks When given a user-specified goal, the robot performs that goal. However, in its "spare time," the robot imagines potential outcomes that it can produce, and then "practices" taking actions to produce them. Each such practice session deepens its understanding of the causal structure in the world.

Of course, the notion of a real-world commercially deployed robotic system that "plays" with its environment in this way might seem far-fetched (it is also of course not a new idea [15]). This is precisely why offline RL is important: since an offline algorithm would be comparatively indifferent to the *source* of the experience, the fraction of time that the robot spends accomplishing user-specified objectives versus "playing" could be adjusted to either extreme, and even a system that spends all of its time performing user-specified tasks can still use all of its collected experience as *offline* training data for learning to achieve any outcome. Such a system would still "play" with its environment, but only virtually, in its "memories."

While robotic systems might be the most obvious domain in which to instantiate this design, it is not restricted to robotics, nor to systems that are embodied in the world in an analogous way to people. Any system that has a well-defined notion of actions can be trained in this way: recommender systems, autonomous vehicles, systems for inventory management and logistics, dialogue systems, and so forth. Online exploration may not be feasible in many of these settings, but learning with unsupervised outcome-driven objectives via offline RL is still possible. As mentioned previously, ML systems are useful insofar as they enable making intelligent decisions. It therefore stands to reason that any useful ML system is situated in a sequential process where decision-making is possible, and therefore such a self-supervised learning procedure should be applicable.

## 3   Unsupervised and Self-Supervised Reinforcement Learning

An unsupervised or self-supervised reinforcement learning method should fulfill two criteria: it should learn behaviors that control the world in meaningful ways, and it should provide some mechanism to learn to control it in as *many* ways as possible. This leaves considerable room for interpretation, for example concerning the term "meaningful," and a wide range of potential definitions have been offered in the literature [16, 17, 18]. This problem should not be confused with the closely related problem of exploration, which has also often been formulated as a problem of attaining broad coverage [19], but which is not generally concerned with learning to control the world in meaningful ways in the absence of a task objective. That is, exploration methods provide an objective for *collecting* data, rather than *utilizing* it.

Perhaps the most direct way to formulate a self-supervised RL objective is to frame it as the problem of reaching a goal state [20]. The problem then corresponds to training a goal-conditioned policy $\pi(\mathbf{a}|\mathbf{s}, \mathbf{g})$ with some choice of reward function $r(\mathbf{s}, \mathbf{g})$. Though this reward itself might constitutes a manually designed objective, it's possible to derive frameworks where the reward function is a consequence of solving a well-defined inference problem, such as the problem of predicting the action that is most likely to lead a particular outcome [21, 22]. This problem formulation provides considerable depth, with connections to density estimation [21], variational inference [23, 22], model-based reinforcement learning [24, 25, 26], and exploration [27].

What does a policy trained to reach all possible goals learn about the world? As argued both in recent work [25, 26] and classic literature in RL [24], solving such goal-conditioned RL problems corresponds to learning a kind of dynamics model. Intuitively, being able to bring about any potential desired outcome requires a deep understanding of how actions affect the environment over a long horizon. Of course, one might then wonder – why not just learn a dynamics model directly, of the form more commonly used in model-based RL? Model learning is likely also an effective way to utilize diverse datasets without a specific user-provided objective. However, there is good reason to believe that goal-conditioned RL either solves a very similar problem [24, 25, 26] or,

potentially, actually solves a problem that is more closely connected to downstream tasks – unlike model-based RL, where the model objective is largely disconnected from actually bringing about desired outcomes, the goal-conditioned RL objective is connected to long-horizon outcomes very directly. Therefore, insofar as the end goal of an ML system is to bring about desired outcomes, we would expect that the objective of goal-conditioned RL would be well-aligned.

However, current approaches suffer from a number of major limitations. Even standard goal-conditioned RL methods can be difficult to use and unstable. But even more importantly, goal-reaching does not span the full set of possible tasks that could be specified in RL. Even if an agent learns to successfully accomplish every outcome that is possible in a given environment, there may not exist a single desired outcome that would maximize an arbitrary user-specified reward function. It may still be that such a goal-conditioned policy would have learned powerful and broadly applicable features, and could be readily finetuned to the downstream task, but an interesting problem for future work is to better understand whether more universal self-supervised objectives could lift this limitation. A number of methods have been proposed for unsupervised acquisition of *skills* [28, 29, 30], so we might reasonably ask whether more general and principled self-supervised reinforcement learning objectives could be derived on this basis.

## 4 Offline Reinforcement Learning

As discussed previously, offline RL can make it possible to apply self-supervised or unsupervised RL methods even in settings where online collection is infeasible, and such methods can serve as one of the most powerful tools for incorporating large and diverse datasets into self-supervised RL. This is likely to be essential to make this a truly viable and general tool for large-scale representation learning. However, offline RL presents a number of challenges [13]. Foremost among these is that offline RL requires answering *counterfactual* questions: given data that shows one outcome, can we predict what would have happened if we had taken a different action? This is of course very challenging in general. Nonetheless, our understanding of offline RL has progressed significantly over the past few years. Besides understanding how distributional shift affects offline RL [31], the performance of offline RL algorithms has advanced considerably, and new algorithms have been developed that provide robustness guarantees [32], finetune online after offline pretraining [14], and tackle a range of other problems in the offline RL setting.

Advances in offline RL have the potential to significantly increase the applicability of self-supervised RL methods. Using the tools of offline RL, it is possible to construct self-supervised RL methods that do not require any exploration on their own. Much like the "virtual play" mentioned in Section 2, we can utilize offline RL in combination with goal-conditioned policies to learn entirely from previously collected data [33, 34, 35]. However, major challenges remain. Offline RL algorithms inherit many of the difficulties of standard (deep) RL methods, including sensitivity to hyperparameters, but these difficulties are further exacerbated by the fact that we cannot perform multiple online trials to determine the best hyperparameters. In supervised learning we can deal with such issues by using a validation set, but a corresponding equivalent in offline RL is lacking. We need algorithms that are more stable and reliable, as well as effective methods for evaluation, in order to make such approaches truly broadly applicable.

## 5 Concluding Remarks

In this article, I discussed how self-supervised reinforcement learning combined with offline RL could enable scalable representation learning. The motivation behind this approach is that, insofar that learned models are useful, it is because they allow us to make decisions that bring about the desired outcome in the world. Therefore, self-supervised training with the goal of bringing about *any* possible outcome should provide such models with the requisite understanding of how the world works. Self-supervised RL objectives, such as those in goal-conditioned RL, have a close relationship with model learning, and fulfilling such objectives is likely to require policies to gain a functional and causal understanding of the environment in which they are situated. However, for such techniques to be useful, it must be possible to apply them at scale to real-world datasets. Offline RL can play this role, because it enables using large, diverse previously collected datasets. Putting these pieces together may lead to a new class of algorithms that can understand the world through action, leading to methods that are truly scalable and automated.

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
