# OpenReview forum: "Understanding the World Through Action"
_robot-learning.org/CoRL/2021/Conference/Blue_Sky — CoRL 2021, Blue Sky_

### Official Review · Reviewer_tYJd · 2021-08-24

**Novelty:** Fair
**Impact:** 3
**Clarity Of Presentation:** Good

**Recommendation:**

Weak Reject: I recommend rejecting the paper, but will not argue for my recommendation if the majority of other reviewers have a different opinion.

**Summary:**

In this blue sky paper, the author proposes that a combination of self-supervised RL and offline RL can help a scalable representation learning. The proposed approach is able to learn the intrinsic mechanisms of the environment and apply them at scale to real-world datasets. Generally, the idea emphasizes learning the environment from action and a feasible approach is provided.

**Summary Of Recommendation:**

The title of this submission is amazing and its motivation is very well. Indeed, learning from action is very important to understand the world. However, the scope is limited to the supervised RL-related method, which is not enough to reflect the nature of learning from action. In addition, embodied learning is rather general but not limited to RL. So I think the scope of this submission is not so general to satisfy the quality of CORL blue sky.

Therefore, I donot recommend to accept this submission as blue sky paper

---

### Official Review · Reviewer_mdMc · 2021-08-25

**Novelty:** Fair
**Impact:** 3
**Clarity Of Presentation:** Good

**Recommendation:**

Weak Accept: I recommend accepting the paper, but will not argue for my recommendation if the majority of other reviewers have a different opinion.

**Summary:**

This paper discusses how unsupervised or self-supervised reinforcement can be developed. The paper first describes the importance of the scalability of datasets and models. Then, the importance of offline RL and self-supervised objectives is explained.


**Summary Of Recommendation:**

The paper covers the relevant recent studies that push the research towards unsupervised/self-supervised RL.  The connection between goal-conditioned RL and self-supervised RL discussed in the paper is interesting. The paper provides a nice overview of the field.

However, considering that many researchers are aware of the importance of self-supervised RL and offline RL,  a new perspective/idea/paradigm is limited.

Overall, I think the paper is a nice short introduction to supervised RL, but the readers should not expect a new perspective/formulation/paradigm.

---

### Decision · Program_Chairs · 2021-10-01

**Decision:**

Accept

**Comment:**

The paper constitutes a nice view on self-supervised reinforcement learning and discusses missing pieces and research challenges connected to it. While one reviewer was concerned w.r.t. the novelty, I think the paper puts together different aspect of the problem nicely and was an interesting read.